# FOXM1-CD44 Signaling Is Critical for the Acquisition of Regorafenib Resistance in Human Liver Cancer Cells

**DOI:** 10.3390/ijms23147782

**Published:** 2022-07-14

**Authors:** Kenly Wuputra, Pi-Jung Hsiao, Wen-Tsan Chang, Po-Hsuan Wu, Lin-Ann Chen, Jian-Wei Huang, Wen-Lung Su, Ya-Han Yang, Deng-Chyang Wu, Kazunari K. Yokoyama, Kung-Kai Kuo

**Affiliations:** 1Graduate Institute of Medicine, Kaohsiung Medical University, Kaohsiung 80708, Taiwan; kenlywu@hotmail.com; 2Regenerative Medicine and Cell Therapy Research Center, Kaohsiung Medical University, Kaohsiung 80708, Taiwan; sal9522059@yahoo.com.tw (Y.-H.Y.); dechwu@kmu.edu.tw (D.-C.W.); 3Cell Therapy and Research Center, Kaohsiung Medical University Hospital, Kaohsiung 80756, Taiwan; 4Department of Internal Medicine, Division of Endocrinology and Metabolism, EDA Hospital, College of Medicine, I-Shou University, Kaohsiung 82445, Taiwan; pjhsiao@gmail.com; 5Center for Cancer Research, Kaohsiung Medical University, Kaohsiung 80708, Taiwan; wtchang@kmu.edu.tw; 6Division of General & Digestive Surgery, Department of Surgery, Kaohsiung Medical University Hospital, Kaohsiung 80756, Taiwan; 1050083@gap.kmu.edu.tw (P.-H.W.); linann423@hotmail.com (L.-A.C.); 7Department of Surgery, School of Medicine, College of Medicine, Kaohsiung Medical University, Kaohsiung 80708, Taiwan; 8Department of Surgery, Ministry of Health and Welfare Pingtung Hospital, Pingtung 900027, Taiwan; 9Department of Surgery, Kaohsiung Municipal Siaogang Hospital, Kaohsiung 812, Taiwan; whalemax@hotmail.com; 10Department of Surgery, Kaohsiung Municipal Ta-Tung Hospital, Kaohsiung 80145, Taiwan; 970401@ms.kmuh.org.tw; 11Division of Gastroenterology, Department of Internal Medicine, Kaohsiung Medical University Hospital, Kaohsiung 80756, Taiwan

**Keywords:** CD44, cancer stem cell lines, FOXM1, hepatocellular carcinoma, regorafenib

## Abstract

Regorafenib is a multikinase inhibitor that was approved by the US Food and Drug administration in 2017. Cancer stem cells (CSCs) are a small subset of cancer-initiating cells that are thought to contribute to therapeutic resistance. The forkhead box protein M1 (FOXM1) plays an important role in the regulation of the stemness of CSCs and mediates resistance to chemotherapy. However, the relationship between FOXM1 and regorafenib resistance in liver cancer cells remains unknown. We found that regorafenib-resistant HepG2 clones overexpressed FOXM1 and various markers of CSCs. Patients with hepatocellular carcinoma also exhibited an upregulation of FOXM1 and resistance to regorafenib, which were correlated with a poor survival rate. We identified a close relationship between FOXM1 expression and regorafenib resistance, which was correlated with the survival of patients with hepatocellular carcinoma. Thus, a strategy that antagonizes FOXM1–CD44 signaling would enhance the therapeutic efficacy of regorafenib in these patients.

## 1. Introduction

Cancer stem cells (CSCs) have the characteristics of extensive proliferation, self-renewal, and increased frequency of tumor formation. Moreover, CSCs can induce the epithelial–mesenchymal transition (EMT), which is responsible for tumor metastasis [1,2]. Distinct populations of CSCs in hepatocellular carcinoma (HCC) have been defined and characterized using normal liver stem cell markers and liver progenitor cell markers, including CD44, CD133/PROM-1, CD90/THY-1, aldehyde dehydrogenase, CD13, oval cell marker OV-6, Sal-like protein 4, CD117/c-kit, intercellular adhesion molecule 1, CD24, delta-like 1, cytokeratin 19, and epithelial cell adhesion molecule (EpCAM) [3,4]. It is reasonable to assume that the transformed marker-negative cancer cells can dedifferentiate to acquire the specific features of liver CSCs [5,6]. In addition, CSCs have been shown to be more resistant to chemotherapy and radiotherapy. We previously reported that the CSC-like cells derived from the HepG2 cell line using OCT4, KLF4, SOX2, and C-MYC (OSKM, Yamanaka factors [7]), together with the *shTP53* knockdown lentivirus, also exhibited drug resistance and upregulation of EMT markers, such as CD44, EpCAM, and CD133 [8]. EpCAM^+^ HCCs are much more sensitive to T cell factor/β-catenin binding inhibitors than EpCAM^−^ HCCs in vitro. Dishevelled-1 is an essential component of the Wnt signaling pathway that stabilizes β-catenin and mediates the Wnt pathway. Dishevelled-1 silencing using small interfering RNA (siRNA) or other small-molecule inhibitors in 5-fluorouracil-resistant HepG2 cells could restore the 5-fluorouracil responsiveness via reduced cell proliferation and migration as well as increased apoptosis [9]. Furthermore, the sorafenib-resistant cell population was enriched for cells with cancer stem-like properties and tumorigenicity, along with high levels of EpCAM, CD133, CD90, and the epithelial progenitor marker cytokeratin 19 [10]. Several liver CSC markers, including LGR5, SOX9, NANOG, and CD90, were elevated in sorafenib-resistant subclones [11]. Drug resistance in cancer is one of the most limiting factors in the clinical treatment of patients with cancer. There are as many underlying mechanisms of resistance as there are patients with cancer because each cancer has its defining set of characteristics that dictates cancer progression and that can eventually lead to death. Therefore, solving this resistance problem seems to be an unattainable goal [12]. Regorafenib was approved by the US Food and Drug Administration (US FDA) in 2017 for the second-line treatment of HCC in adult patients who had previously received sorafenib therapy [13,14,15,16] or experienced disease progression after cancer treatment [17]. Despite the survival benefit of regorafenib for patients with metastatic cancer, its overall clinical efficacy remains limited. Thus, the identification of the target genes involved in the drug resistance induced by regorafenib and their mechanistic studies might be crucial for the development of new treatments for HCC.

The forkhead box protein M1 (FOXM1) is a member of the FOX transcription factor family and plays an important role in the regulation of cell-cycle progression, drug resistance, CSC renewal, and cancer differentiation [18,19]. A recent study demonstrated that the upregulation of FOXM1 was associated with a poor prognosis among patients with HCC, colorectal cancer, and lung cancer [20]. Thus, FOXM1 was regarded as a key transcription factor associated with HCC [21]. Conversely, inhibition of FOXM1 in liver cancer cells reduced cell proliferation, angiogenesis, and EMT [22]. These findings indicate the critical role of FOXM1 in tumorigenesis and cancer development. However, the mechanism underlying FOXM1 dysregulation remains to be determined. Some studies demonstrated that the oncogenic activity of FOXM1 can be regulated, at least in part, by the PI3K/AKT and/or RAS/MAPK signaling pathways [19,23].

Based on these observations, we focused our analyses on regorafenib resistance and its underlying mechanisms, such as the examination of target genes, including FOXM1 and the CSC marker CD44. Thus, we examined the role of FOXM1 in the establishment of the CSC characteristics and regorafenib resistance via CD44 using the HepG2 and Hep3B liver cancer cell lines.

## 2. Results

### 2.1. FOXM1 and Various CSC Markers Were Significantly Overexpressed in Regorafenib-Resistant Cells

HepG2_Rego_R cells were generated by continuous cultivation using 2–6 µM regorafenib for 6 months. Subsequently, the cells became loosely attached to the substratum but remained attached to each cell, and then they were aggregated to form the spheroid structures (Figure 1A). To examine the cell viabilities of HepG2, HepG2_Rego_R, Hep3B, and Hep3B_Rego_R cells, 2–10 µM of regorafenib was used to calculate the half-maximal inhibitory concentration (IC50; Appendix A). The IC_50_ values of HepG2 and HepG2_Rego_R cells were 7.3 µM and 10 µM, respectively. Thus, we determined that the suitable concentration of regorafenib for further experiments was 5 µM. The HepG2_Rego_R cells exhibited a 3.75-fold higher colony-forming ability than the parental HepG2 cells in the absence of regorafenib. In the presence of 5 µM regorafenib, the colony number for the HepG2_Rego_R cells increased by 1.45-fold, but their colony size seemed to be smaller than that of the dimethyl sulfoxide (DMSO) control cells, without 5 µM regorafenib treatment. In the case of HepG2 cells, no colonies were obtained after regorafenib treatment (Figure 1B). However, even after treatment with 5 µM regorafenib, larger spheroids were obtained for HepG2_Rego_R cells than those obtained for HepG2 cells without 5 µM regorafenib treatment. Hep3B_Rego_R cells were also obtained using the same conditions with the addition of 5 µM regorafenib. The colony size and number characteristics of these clones were similar to those of the HepG2_Rego_R cells, i.e., smaller colony sizes and approximately twofold higher colony numbers than those for the parental cell colonies. Spheroids of the HepG2_Rego_R cells were 2–5-fold larger than those of the HepG2 cells both in the absence and presence of regorafenib (Figure 1C). Larger spheroids were obtained for the HepG2_Rego_R cells than for HepG2 cells, even on treatment with 5 µM regorafenib. Similar observations were made for the Hep3B_Rego_R cells (data unpublished).

Next, we examined the expression levels of stem cell and CSC markers. The mRNA expressions of stem cell markers, such as ATP binding cassette subfamily G member 2 (ABCG2), SOX2, GATA-binding factor 6, and CD44, were significantly increased in HepG2_Rego_R cells compared with HepG2 cells (Figure 2A) [24,25]. These marker proteins were also confirmed to express higher (Figure 2B).

We also examined FOXM1, which is a transcription factor that plays an important role in the cell cycle [18]. Recently, FOXM1 was also reported as an oncogene in a variety of cancers, including lung cancer [26], breast cancer [27], colorectal cancer [28], and HCC [21,29]. The expression of FOXM1 was dramatically increased in HepG2_Rego_R cells compared with that in HepG2 cells. Moreover, the RNA expression of the possible downstream target genes of FOXM1, including Aurora kinase A (AURKA) [30] and BIRC5/Survivin [31], was also significantly increased (3–14-fold; Figure 2C) in the HepG2_Rego_R cells. In addition, we examined the protein expression levels of FOXM1, AURKA, and BIRC5 and found that they were higher in the HepG2_Rego_R cells (approximately 2–18-fold) than in HepG2 cells (Figure 2D). Taken together, these findings indicate that the regorafenib-resistant cancer cells exhibited characteristics more similar to those of CSCs, with a higher expression of stem cell and CSC markers.

### 2.2. The EMT Phenomenon Occurred in Regorafenib-Resistant Cells

EMT has been reported as one of the tumor-related functions that might enable the CSCs to gain the abilities of self-renewal and invasion and prevent apoptotic activity [32,33,34]. To understand the EMT phenomenon in regorafenib-resistant cells, we measured the expression of EMT marker genes using reverse transcription-quantitative polymerase chain reaction (RT-qPCR) (Figure 3A). The increased expression levels of vimentin (VIMENTIN), twist family BHLH transcription factor 1 (TWIST1), and zinc finger E-box binding homeobox 1 (ZEB1) were detected in the HepG2_Rego_R cells as compared with the control HepG2 cells. Western blotting of EMT-related proteins, such as TWIST1, VIMENTIN, ZEB1, cadherin 1 (CDH1), cytokeratin 7 (CK7), and cytokeratin 18 (CK18), was also performed. In general, three genes, TWIST1, VIM, and ZEB1, have been reported as mesenchymal cell markers, which resulted in the induction of alterations of cell morphology, motility, and adhesion ability [35,36,37,38,39]. In addition, CDH1, CK18, and CK7 are considered epithelial cell markers, and the downregulation of these three genes would force the cells into EMT [40,41,42].

Both HepG2_Rego_R and Hep3B_Rego_R cells exhibited EMT characteristics, such as elevated expressions of TWIST1/2, VIMENTIN, and ZEB1, together with low expressions of epithelial marker proteins, such as CDH1, CK18, and CK7 (Figure 3B). The transwell migration assay demonstrated that the migration ability of HepG2_Rego_R cells was at least twofold greater than that of the parental HepG2 cells (Figure 3C). These findings suggest that the regorafenib-resistant liver cancer cells are more malignant than the parental liver cancer cells.

### 2.3. Inhibition of FOXM1 Restored Cell Death in Regorafenib-Resistant Cells

To understand the role of FOXM1 in cancer progression and drug resistance, we used the FOXM1 inhibitor thiostrepton to determine the role of FOXM1 in cell death and colony formation [43]. The relative expression levels of stem cell and CSC markers were analyzed using Western blotting. Treatment of HepG2_Rego_R cells with 1 µM thiostrepton for 72 h resulted in a significant reduction in protein and RNA expressions of CSC-related markers, such as SOX2, CD44, and BIRC5, as well as control FOXM1 levels (Figure 4A,B). However, the expression trends of some EMT markers were variable. The expression levels of TWIST1/2 were decreased, whereas those of both CK18 and CDH1 were increased after exposure to thiostrepton (Figure 4A). Thus, EMT markers might show different expression patterns indicating the heterogeneity of the EMT sensitivities of TWIST1/2 and other markers, such as CDH1 and CK18 (Figure 4A,B). In addition, the cell viability was reduced in a dose-dependent manner after exposure to thiostrepton for 48 h (Figure 4C) and 72 h (Figure 4D). 

The viabilities of HepG2_Rego_R and Hep3B_Rego_R cells on treatment with 1 µM of thiostrepton were not significantly changed at 3 and 6 days after the treatment compared with those of HepG2 and Hep3B parental cells subjected to the same treatment (Appendix A). Thus, drug-resistant clones, such as HepG2_Rego_R and Hep3B_Rego_R cells, did not show severe cytotoxicity on thiostrepton treatment. 

Moreover, the size of spheres and the sphere-forming ability of the HepG2_Rego_R and Hep3B_Rego_R cells were decreased after FOXM1 inhibition (Figure 5A,B). Furthermore, the colony-forming abilities of the HepG2_Rego_R and HepG2 cells were greatly decreased (by 20–55%) after treatment with 0.5 µM thiostrepton compared with the control DMSO-treated cells (Appendix A). Thus, the effects of 0.5 µM thiostrepton on the Hep3B_Rego_R and Hep3B cells were similar to those on the HepG2_Rego_R and HepG2 cells (unpublished data).

### 2.4. FOXM1 Knockdown Impaired CD44 and SOX2 Expression and the CSC Population in Regorafenib-Resistant Cells 

We performed knockdown experiments using short hairpin RNA (shRNA; *shFOXM1*) constructs targeting FOXM1 and its control off-target construct in HepG2_Rego_R and Hep3B_Rego_R cells. The expression of the CSC markers SOX2 and CD44 was significantly decreased in protein levels. In addition, the expression of AURKA in the knockdown cells was also decreased by 50% compared with those in the control HepG2_Rego_R and Hep3B_Rego_R cells (Figure 6A). This effect might be regulated by nutrient availability, as reported previously [44,45]. Thus, we then examined the role of FOXM1 in CSCs under the condition of nutrient deprivation by culturing HepG2_Rego_R and Hep3B_Rego_R cells, their parental cells, and the *shFOXM1*-transfected cells in Dulbecco’s modified Eagle’s medium (DMEM) containing 1% fetal bovine serum (Appendix A). Compared with their parental cells, HepG2_Rego_R and Hep3B_Rego_R cells exhibited a higher survival. However, the introduction of *shFOXM1* resulted in a decrease in the survival of HepG2_Rego_R and Hep3B_Rego_R cells. Moreover, the size of spheres and the sphere-forming ability of HepG2_Rego_R and Hep3B_Rego_R cells were decreased significantly after the introduction of the *shFOXM1* construct (Figure 6B,C). In the case of *siRNA-FOXM1* use, the protein expression of SOX2 and CD44 was completely knocked down, which was similar to the effects of shRNA-FOXM1 (data not shown). 

Because CD44 expression is highly correlated with FOXM1 expression [6], we predicted that FOXM1 would play a critical role in the promoter activity of the *CD44* gene. Thus, we performed a *CD44* promoter-driven luciferase assay. The *CD44* promoter activity was dramatically increased in HepG2_Rego_R cells compared with that in the parental HepG2 cells, although the control shRNA against the off-targets of FOXM1 did not show any significant *CD44* promoter activity reduction. Thus, the knockdown of FOXM1 resulted in the reduction in *CD44* promoter-luciferase activity (Figure 7). Taken together, these findings indicate that FOXM1 knockdown impaired the expression of CD44 and SOX2 as well as cell proliferation and sphere formation as CSC features in regorafenib-resistant cells.

### 2.5. FOXM1 Overexpression Was Correlated with Poor Prognosis and Tumor Growth in Patients with HCC 

To investigate the FOXM1 signaling pathway and its possible downstream proteins, including AURKA and BIRC5, in patients with HCC, the whole cohort of patients with HCC and those with Hepatitis virus infection as well as HCC were divided into high- and low-expression groups. Using The Cancer Genome Atlas database, we found that the patient group with higher expressions of FOXM1, AURKA, or BIRC5 exhibited a shorter overall survival than did the group with lower expressions of these proteins, regardless of the hepatitis virus infection status (Figure 8A). In addition, we found that the expression of CSC and EMT markers, such as CD44, SOX2, ABCG2, and VIMENTIN, was closely associated with the survival of patients with HCC. As with the observation for FOXM1, the total cohort of patients with HCC and higher expression of CD44, SOX2, ABCG2, and VIMENTIN exhibited a poorer prognosis than patients with HCC and lower expressions of these proteins (Figure 8B,C). However, among the patients with hepatitis virus infection and HCC, those with higher expressions of ABCG2 and VIMENTIN showed a longer survival than those with lower expressions of these proteins (Figure 8B,C).

Finally, a xenotransplantation experiment was performed to analyze the role of FOXM1 in tumor progression (Figure 9A). HepG2 and HepG2_Rego_R cells treated with or without thiostrepton or 0.05% DMSO control for 72 h were injected subcutaneously into severe combined immunodeficiency mice. Tumor weight in the HepG2_Rego_R mice group was approximately 10-fold higher than that in the HepG2 mice group (Figure 9B). Furthermore, necrosis, number of giant cells, and abnormal mitosis appeared to be significant compared with the control HepG2 group (Figure 9C). Moreover, the tumor weight in the thiostrepton-treated HepG2_Rego_R group was approximately 60% lower than that in the untreated HepG2_Rego_R group. These findings suggest that FOXM1 plays an important role in the progression of HCC and affects the survival rate of patients with HCC.

## 3. Discussion

We previously generated CSC-like cells from the HepG2 cell line using OCT4, KLF4, SOX2, and C-MYC (OSKM), together with the *shTP53* knockdown lentivirus, which also exhibited drug resistance and upregulation of EMT markers, such as CD44, EpCAM, and CD133 [8]. Drug resistance during cancer treatment is considered a limiting factor in curing patients with cancer [12]. Multiple signaling pathways and mechanisms are involved in the development of drug resistance during the treatment of various cancers, including HCC. Some studies indicated that resistance to tyrosine kinase inhibitors (TKIs) could enrich the CSC characteristics and subsequently lead to drug resistance [46]. A TKI is designed to inhibit the corresponding kinase from exerting its function of catalyzing phosphorylation [47]. Since the US FDA approved imatinib for the treatment of chronic myeloid leukemia in 2001, multiple potent and well-tolerated TKIs—with targets including EGFR, ALK, ROS1, HER2, NTRK, VEGFR, RET, MET, MEK, FGFR, PDGFR, and KIT—have emerged and contributed to significant progress in cancer treatment. Regorafenib was approved by the US FDA to treat various cancers, including HCC [48]. However, its efficacy has been restricted by the emergence of TKI resistance.

In the present study, both regorafenib-resistant HepG2 and Hep3B cells exhibited overexpressions of FOXM1 as well as CSC and EMT markers. FOXM1 plays a major role in the progression of various cancers [49]. Moreover, the FOXM1 pathway is involved in TKI resistance in lung cancer [50]. However, the role of FOXM1 in TKI resistance has not been studied in HCC.

Our present in vitro and in vivo analyses showed that CSC and EMT markers and FOXM1 were overexpressed in regorafenib-resistant HepG2 cells. FOXM1 and downstream molecules, such as BIRC5 (survivin), may serve as predictive factors and therapeutic targets in HCC. First, the efficient downregulation of FOXM1 in regorafenib-resistant HepG2 cells reduced cell viability, sphere formation, and colony formation. Second, the downregulation of FOXM1 using thiostrepton (a FOXM1 inhibitor) or shRNA impaired CSC phenotypic characteristics, such as SOX2 and CD44 downregulation, in regorafenib-resistant HepG2 cells. Third, the inhibition of FOXM1 significantly decreased tumor growth in mouse xenografts. Importantly, we found that the overexpression of FOXM1 was correlated with an upregulation of CD44 and SOX2 and was associated with a poorer prognosis in patients with HCC. Our findings demonstrate a novel molecular mechanism for regorafenib resistance in HepG2 and Hep3B cells and provide potential predictive factors and therapeutic targets for future clinical applications in human HCC.

The upregulation of FOXM1 was detected in regorafenib-resistant HepG2 and Hep3B cells and led to the enhancement of CSC ability and the induction of EMT (Figure 1, Figure 2 and Figure 3). Moreover, the downregulation of FOXM1 activity using an inhibitor or shRNA increased cell death and decreased cell proliferation, the size of spheres, and the sphere-forming ability in regorafenib-resistant HepG2 and Hep3B cells (Figure 4 and Figure 5) [26]. The knockdown of FOXM1 downregulated the CD44 and SOX2 CSC markers (Figure 6), which are involved in the emergence of CSCs in various types of cancers [50]. It has been reported that FOXM1 could directly upregulate CD44 and trigger stem cell features, to enhance the progression of and cell survival in RAS-driven HCC [51]. Using the *CD44* promoter-luciferase assay, we demonstrated that the knockdown of FOXM1 decreased the expression of CD44 in regorafenib-resistant HepG2 and Hep3B cells (Figure 7). This reduction caused by *shFOXM1* might be due to the binding of FOXM1 to site 4714 in the *CD44* promoter region because the chromatin immunoprecipitation (ChIP) assay demonstrated that FOXM1 appeared to be recruited to this site (Appendix A). Further experiments will be required to conclude whether FOXM1 binds to site 4714 in the *CD44* promoter to transactivate this promoter. Taken together, our observations suggest that FOXM1 plays an important role in regorafenib resistance, which is mediated via the overexpression of CD44.

In patients with HCC and a higher expression of FOXM1, we also detected high expressions of CD44, SOX2, ABCG2, and VIMENTIN, as well as a poor prognosis (Figure 8A–C). Previous studies reported that AURKA activity was mediated by FOXM1 activation for regulating mitotic-spindle assembly and chromosome segregation [46]. Moreover, BIRC was found to be induced after FOXM1 activation of the G2/M checkpoint [47]. Consistent with our in vivo findings, a poor prognosis and short overall survival rate were observed for patients with HCC and high expressions of FOXM1 and its downstream proteins, AURKA and BIRC5 (Figure 8A,B). In the in vivo model, mice transplanted with thiostrepton-treated regorafenib-resistant HepG2 cells showed smaller sizes of the xenotransplanted tumors derived from regorafenib-resistant HepG2 cells than did those transplanted with untreated regorafenib-resistant HepG2 cells (Figure 9A,B). To demonstrate the role of FOXM1 in vivo, we need to examine the effect of the genetically induced depletion of the FOXM1 gene or its inhibitor in the liver from the regorafenib-resistant mice model.

In summary, FOXM1 was overexpressed in regorafenib-resistant HepG2 and Hep3B cells, both of which also showed upregulations of CSC and EMT markers in vivo and in vitro. The inhibition of FOXM1 restored TKI sensitivity and cell death and reduced sphere formation in regorafenib-resistant HepG2 and Hep3B cells. Furthermore, the downregulation of FOXM1 activity using a FOXM1 inhibitor or shRNA impaired CD44 and SOX2 expression. In a mouse xenograft study, engrafted tumor cells with a low expression of FOXM1 led to a reduced tumor size of the xenograft when harvested. Our findings enrich our knowledge about FOXM1, and its downstream genetic networks involved in regorafenib resistance. The present study not only identified the possible mechanisms underlying TKI resistance but also indicates that players in the FOXM1–CD44 network may become promising therapeutic targets to reverse TKI resistance in HCC.

## 4. Materials and Methods

### 4.1. Cells

The human HCC cell lines HepG2 and Hep3B were prepared as described previously [52]. The regorafenib-resistant HepG2_Rego_R and Hep3B_Rego_R cell lines were generated by culturing cells with regorafenib, starting at the concentration of 2 µM and increasing it up to 6 µM for at least 6 months. Subsequently, the cells were routinely maintained at the 6 µM concentration for each experiment.

### 4.2. Plasmid DNA, siRNA, and shRNA Transfections

Cells (5 × 10^6^) were transiently transfected with plasmid DNAs, siRNA, and shRNA for FOXM1 using Lipofectamine 2000 (Thermo Fisher Scientific, Waltham, MA, USA) according to the manufacturer’s instructions. After incubation for 48 h, the cells were collected and harvested for subsequent experiments. After shRNA transfection for 48 h, we selected the best clone for further analysis, as assessed by the significant reduction in FOXM1 expression, which was evaluated using Western blotting and RT-qPCR (Table 1).

The siRNA for FOXM1 and the corresponding control nontarget siRNA were purchased from Horizon Discovery (Level Biotechnology, Inc., New Taipei, Taiwan; D-001810-10-05 and L-009762-00-0005). shRNAs were purchased from RNAi core, Academia Sinica, Taipei, Taiwan (TRCN0000273981 and ASN0000000003).

### 4.3. Cell Viability

Cells (5 × 10^3^) in 96-well plates were treated with the indicated concentrations of regorafenib (Stivarga^®^, Bayer Pharma AG, Berlin, Germany) and thiostrepton (T8902, Sigma-Aldrich, St. Louis, MO, USA) for 72 h. Then, cell viability was examined using the 3-(4,5-dimethylthiazolyl-2)-2,5-diphenyltetrazolium bromide assay (0.5 mg/mL) according to the manufacturer’s instructions. The half-maximal inhibitory concentration of the drugs was also determined.

### 4.4. Dual-Luciferase Assay

Cells (5 × 10^6^) were transiently transfected with the CD44P pGL3 plasmid (19122; Addgene, Watertown, MA, USA) and the pRL-CMV plasmid encoding Renilla luciferase using Lipofectamine 2000 (Invitrogen, Waltham, MA, USA) according to the manufacturer’s instructions. After incubation for 48 h, the cells were harvested and the luciferase activities were measured using the Dual-Luciferase^®^ Reporter Assay System (Promega, Madison, WI, USA) according to the manufacturer’s instructions.

### 4.5. 3-D-Sphere- and Colony-Formation Assays

Sphere- and colony-formation assays were performed as described previously [52]. Briefly, for sphere formation, cells were grown in serum-free DMEM (HyClone, Cytiva, Tokyo, Japan) supplemented with B-27 (Invitrogen), 20 ng/mL of epidermal growth factor, and 20 ng/mL of basic fibroblast growth factor (ProSpec-Tany TechnoGene Ltd., Rehovot, Israel). Cells were then plated in 6-well ultra-low-attachment plates (Corning, Glendale, AZ, USA) and sphere formation was assessed using a microscope after 6 days of growth. For the colony-formation assay, cells were plated in a gelatin-coated dish at a density of 5 × 10^2^ cells. Two weeks later, colonies with a diameter >2 mm were counted after staining with Giemsa staining solution (Wako Chemicals, Tokyo, Japan).

### 4.6. Migration Assay

Cells were seeded on a transwell plate and incubated without serum. The transwell was then placed on a plate containing DMEM with 10% fetal bovine serum for 48 h. The migrated cells located on the lower surface of the filters were fixed, stained, and counted using a microscope.

### 4.7. Western Blot Analysis

Western blot analysis was performed as described previously [52,53]. The following antibodies were used: anti-FOXM1 (GTX100276), anti-BIRC5 (GTX100441), anti-AURKA (GTX13824), anti-CD44 (GTX102111), anti-TWIST1/2 (GTX127310), anti-ZEB1 (GTX105278), anti-CDH1 (GTX02618), anti-CK18 (GTX105624), anti-CK7 (GTX109723), and anti-VIMENTIN (GTX100619), which were obtained from GeneTex, Hsinchu City, Taiwan; anti-GAPDH (MAB374) and anti-SOX2 (AB5603), which were obtained from Merck Millipore (Burlington, MA, USA).

### 4.8. In Vivo Tumor Xenograft Model

HepG2 cells, HepG2_Rego_R cells, and their FOXM1 inhibitor or control 0.05% DMSO-treated (for 72 h) counterparts (1 × 10^6^) were injected subcutaneously into severe combined immunodeficiency mice (male, 8 weeks; the National Laboratory Animal Center (NLAC), Taipei, Taiwan). Tumor size was calculated according to the following formula: (1)$$\mathrm{tumor}\;\mathrm{volume}=\frac{\left(\mathrm{length}\times {\mathrm{width}}^{2}\right)}{2}$$

Four weeks later, the mice were sacrificed, and the tumors were isolated, subjected to fixation in 4% paraformaldehyde, embedded in paraffin, and then sectioned for hematoxylin and eosin staining. All animal experiments were performed in accordance with the animal welfare guidelines for the care and use of laboratory animals published by the NLAC and Kaohsiung Medical University (KMU 106226) in Taiwan.

### 4.9. Statistical Analyses

Data are presented as mean ± SEM from triplicate experiments and additional replicates, as indicated. One-way ANOVA (*p* < 0.0001) followed by two-tailed Student’s *t*-tests was used to assess statistical significance. Survival analysis was performed using the Kaplan–Meier method, and the curves were compared using the log-rank test. *p* < 0.05 was considered statistically significant.

## 5. Conclusions

FOXM1 directly upregulated CD44 expression and triggered stem cell features, to enhance the progression of HCC and cell survival. According to our findings, FOXM1 and CD44 were upregulated in the HepG2_Rego_R cells. Thus, FOXM1 seems to be involved in drug resistance, such as that to regorafenib, through CD44.

## Figures and Tables

**Figure 1 ijms-23-07782-f001:**
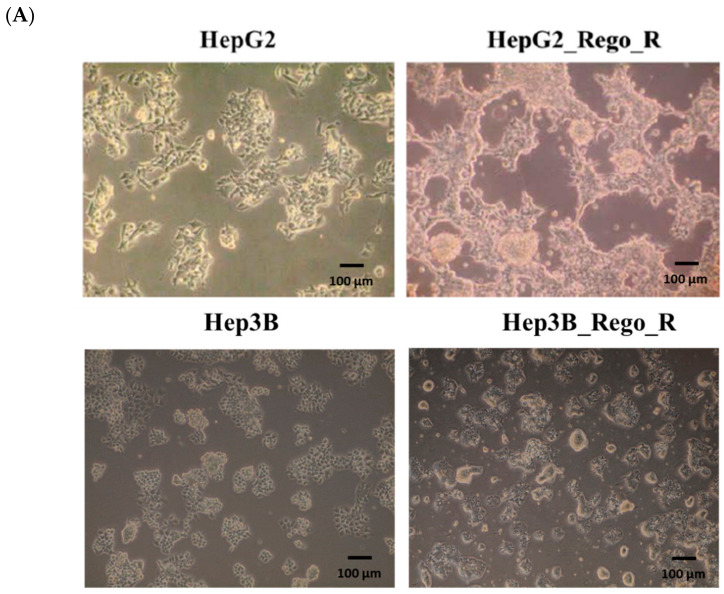
Cell morphologies, colony-forming abilities, and sphere-forming abilities of HepG2 and Hep3B parental cells and their regorafenib-resistant counterparts. (**A**) Cell morphologies of HepG2 cells (upper left panel), Hep3B cells (lower left panel), and regorafenib-resistant HepG2_Rego_R cells (upper right panel) and Hep3B_Rego_R cells (lower right panel). (**B**) Colony-forming abilities of HepG2 cells (upper panel) and HepG2_Rego_R cells (lower panel), and Hep3B cells (upper panel) and Hep3B_Rego_R cells (lower panel). Quantification of the colony number for each cell and the parental cells (right panels) was performed as described in Materials and Methods. Data are presented as mean ± standard error of the mean (SEM; n = 3) and were analyzed using one-way analysis of variance (ANOVA) and the Tukey post hoc test (* *p* < 0.05, ** *p* < 0.01 and *** *p* < 0.001). (**C**) Morphology of spheroids of HepG2 cells (upper panel) and HepG2_Rego_R cells (lower panel) after cultivation for 2 and 6 days without regorafenib and 6 days with 5 µM regorafenib.

**Figure 2 ijms-23-07782-f002:**
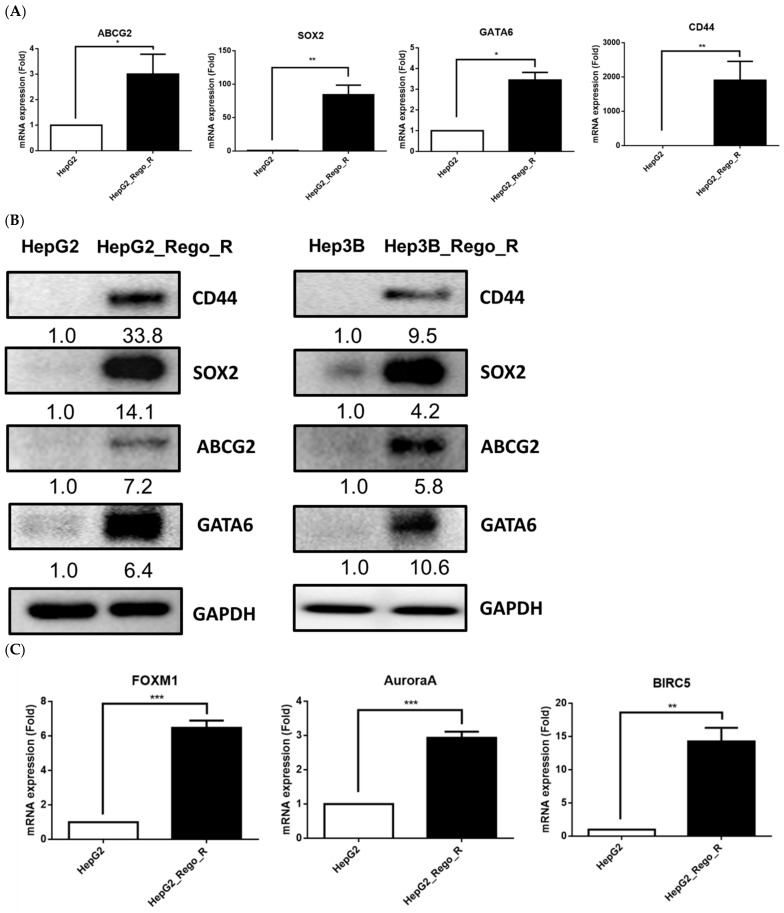
Comparative expression of stem cell and CSC markers between HepG2 and HepG2_Rego_R cells. (**A**) Relative RNA expression levels of ABCG2, SOX2, GATA6, and CD44 using reverse transcription-quantitative polymerase chain reaction. The RNA level in HepG2 cells was regarded as 1.0 and the relative RNA expression in HepG2_Rego_R cells was calculated. (**B**) Protein expression levels of CD44, SOX2, ABCG2, and GATA6 were compared between the HepG2 and HepG2_Rego_R cells, and Hep3B and Hep3B_Rego_R cells using Western blotting. The relative protein expression levels were calculated based on the protein levels in HepG2 and HepG3 cells. (**C**) RNA expression levels of FOXM1, AURKA, and BIRC5 were compared between HepG2 and HepG2_Rego_R cells. (**D**) Comparative protein expression levels of FOXM1, AURKA, and BIRC5 between HepG2 and HepG2_Rego_R cells. The relative protein expression levels were calculated based on the protein levels in HepG2 cells. Statistical analysis was performed as described in Materials and Methods. Data are presented as mean ± SEM (n = 5) and were analyzed using two-way ANOVA and Bonferroni posttests (* *p* < 0.05, ** *p* < 0.01, and *** *p* < 0.001).

**Figure 3 ijms-23-07782-f003:**
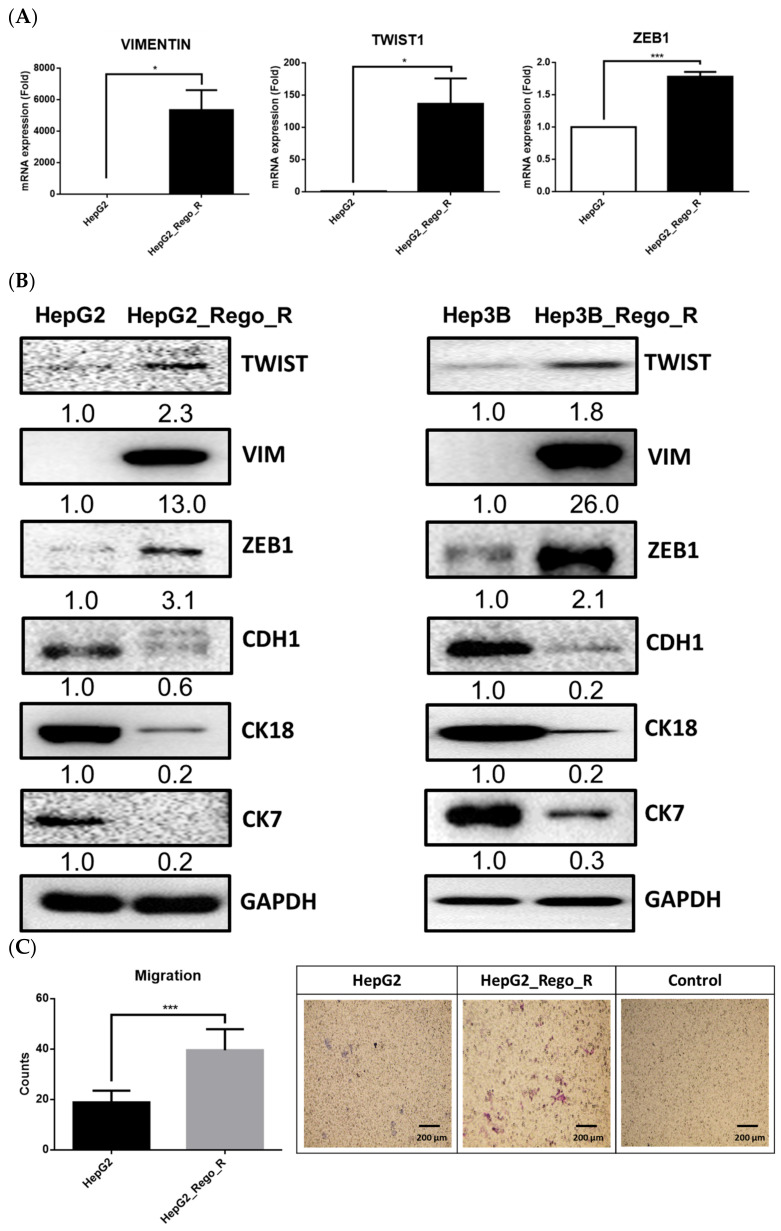
Relative expression of EMT-related markers between HepG2 and HepG2_Rego_R cells. (**A**) RNA expression levels of VIMENTIN, TWIST1, and ZEB1 were determined using RT-qPCR. Data are presented as mean ± SEM (n = 5) and were analyzed using two-way ANOVA and Bonferroni posttests (* *p* < 0.05, *** *p* < 0.001). (**B**) Comparative expression of EMT-related proteins between HepG2 and HepG2_Rego_R cells as well as Hep3B and Hep3B_Rego_R cells. The expression levels of HepG2 and Hep3B cells are regarded as 1.0. Data are presented as mean ± SEM (n = 3) and were analyzed using Student’s *t*-test. (**C**) The migration abilities of HepG2 and HepG2_Rego_R cells were compared as described in Materials and Methods. One of the representative results and control HepG2_Rego_R cells before migration at 0 h is presented in the right panel. The data are presented as mean ± SEM (n = 3) and were analyzed using Student’s *t*-test (*** *p* < 0.001).

**Figure 4 ijms-23-07782-f004:**
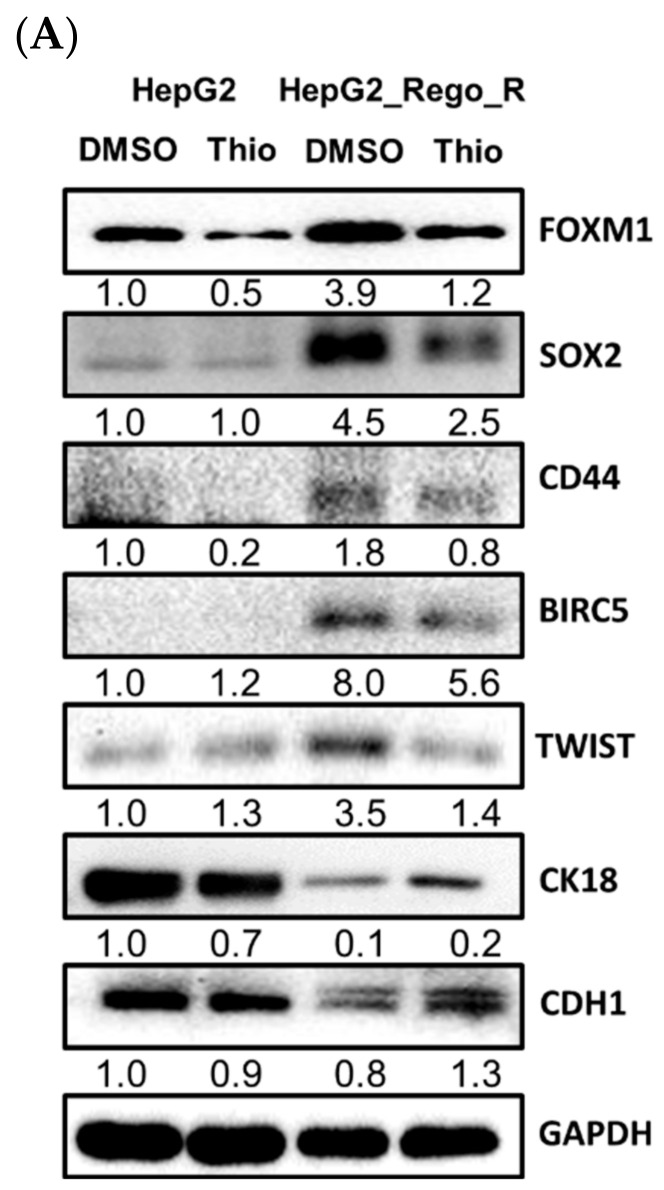
Effects of addition of FOXM1 inhibitor thiostrepton on the expression of FOXM1 signaling pathway-related proteins and cell viability in HepG2_Rego_R cells. (**A**) The comparative expression levels of FOXM1 signaling pathway-related proteins in HepG2 and HepG2_Rego_R cells treated with thiostrepton for 72 h were analyzed using Western blotting as described in Materials and Methods. (**B**) RNA expression levels of FOXM1, BIRC5, and CD44 in HepG2 and HepG2_Rego_R cells treated with thiostrepton were analyzed using RT-qPCR as described in Materials and Methods. Data are presented as mean ± SEM (n = 5) and were analyzed using two-way ANOVA with Bonferroni posttests (*** *p* < 0.001). (**C**,**D**) Cell viabilities of HepG2 and HepG2_Rego_R cells after treatment with thiostrepton for 48 h (**C**) and 72 h (**D**) were measured as described in Materials and Methods. Data are presented as mean ± SEM (n = 5) and were analyzed using Student’s *t*-test (*** *p* < 0.001).

**Figure 5 ijms-23-07782-f005:**
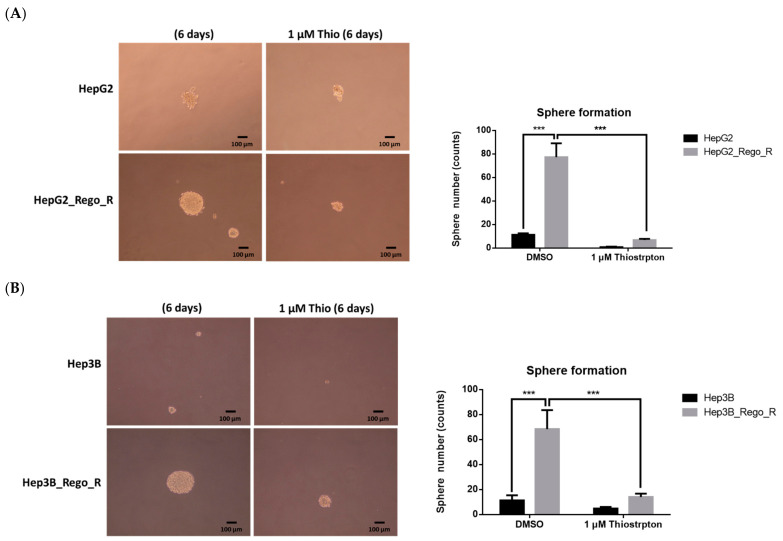
Sphere-forming abilities of HepG2 and HepG2_Rego_R cells, as well as Hep3B and Hep3B_Rego_R cells. HepG2 and HepG2_Rego_R cells (**A**), as well as Hep3B and Hep3B_Rego_R cells (**B**), were treated with 1 µM thiostrepton for 6 days and their sphere-forming abilities were measured as described in Materials and Methods (n = 4). Statistical analysis was performed using two-way ANOVA with Bonferroni posttests (*** *p* < 0.001).

**Figure 6 ijms-23-07782-f006:**
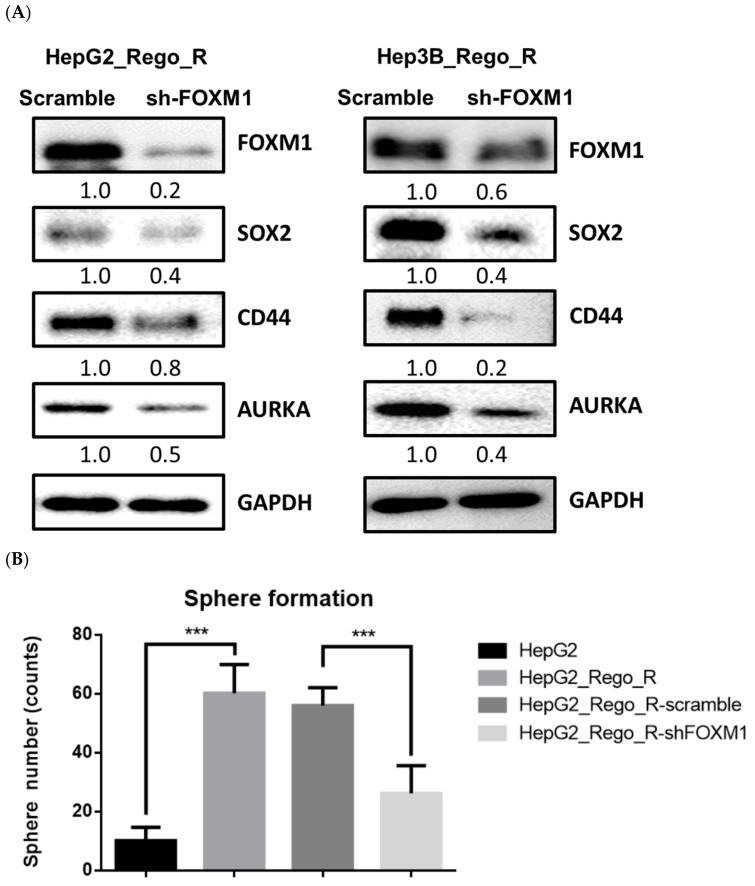
Knockdown of FOXM1 impaired the expression of CD44 and SOX2 and sphere formation in HepG2_Rego_R and Hep3B_Rego_R cells. (**A**) The expressions of FOXM1 and CSC-related proteins, such as SOX2 and CD44, in HepG2_Rego_R (left panel) and Hep3B_Rego_R cells (right panel) were compared after the introduction of *shFOXM1* as described in Materials and Methods. The off-target shRNA was also introduced as the control as described in Materials and Methods. Relative expression levels were calculated based on the expression levels in the cells treated with the control shRNA. (**B**,**C**) Sphere-forming abilities of HepG2_Rego_R (**B**) and Hep3B_Rego_R cells (**C**) after treatment with *shFOXM1* as described in Materials and Methods. The viabilities of the off-target shRNA-treated HepG2_Rego_R and Hep3B_Rego_R cells were not changed. Data are presented as mean ± SEM (n = 5) and were analyzed using one-way ANOVA and the Tukey post hoc test (*** *p* < 0.001).

**Figure 7 ijms-23-07782-f007:**
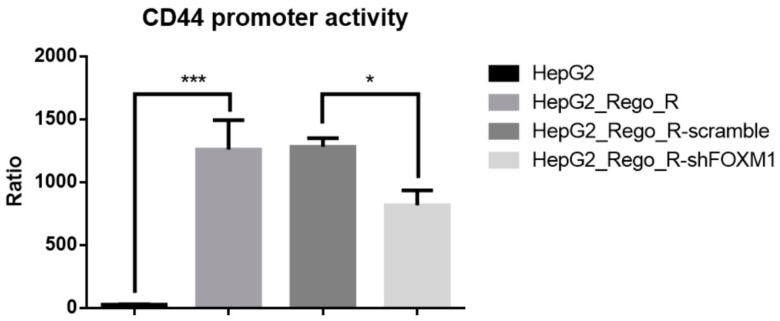
Comparative *CD44*-promoter activities of HepG2 and HepG2_Rego_R cells. Promoter activity of the *CD44* gene in HepG2_Rego_R cells was significantly enhanced, and on using shRNA against FOXM1, it decreased by approximately 65%. The *CD44* promoter activity was not changed in the off-target shRNA-treated HepG2_Rego_R cells compared with that in the control HepG2_Rego_R cells. Data are presented as mean ± SEM (n = 4) and were analyzed using one-way ANOVA and the Tukey post hoc test (* *p* < 0.05 and *** *p* < 0.001).

**Figure 8 ijms-23-07782-f008:**
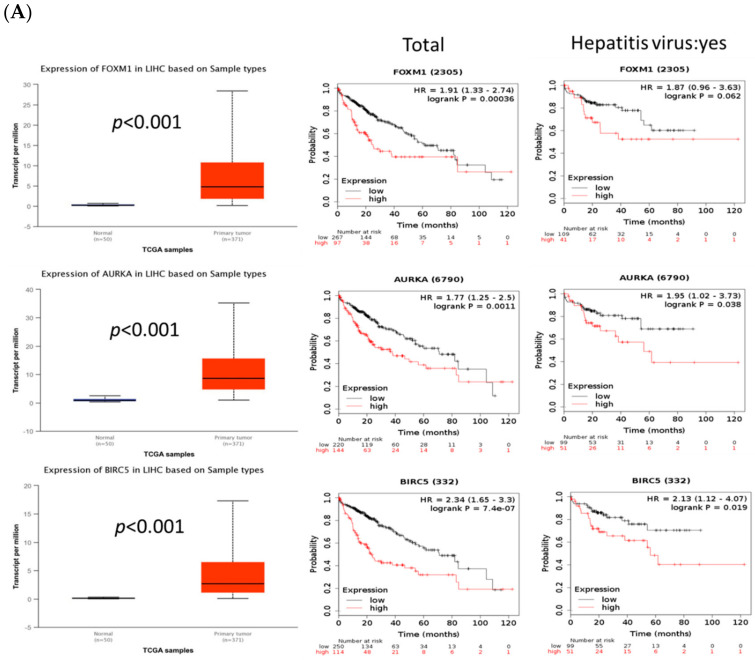
Increased FOXM1 expression was correlated with poor prognosis in patients with HCC and higher tumor growth in mice xenotransplantation experiments. (**A**–**C**) Kaplan–Meier survival curves were estimated using the expression levels of FOXM1, AURKA, BIRC5, CD44, SOX2, ABCG2, and VIMENTIN in The Cancer Genome Atlas database for the whole cohort of patients with HCC (n = 364) and those with hepatitis virus infection as well as HCC (n = 150).

**Figure 9 ijms-23-07782-f009:**
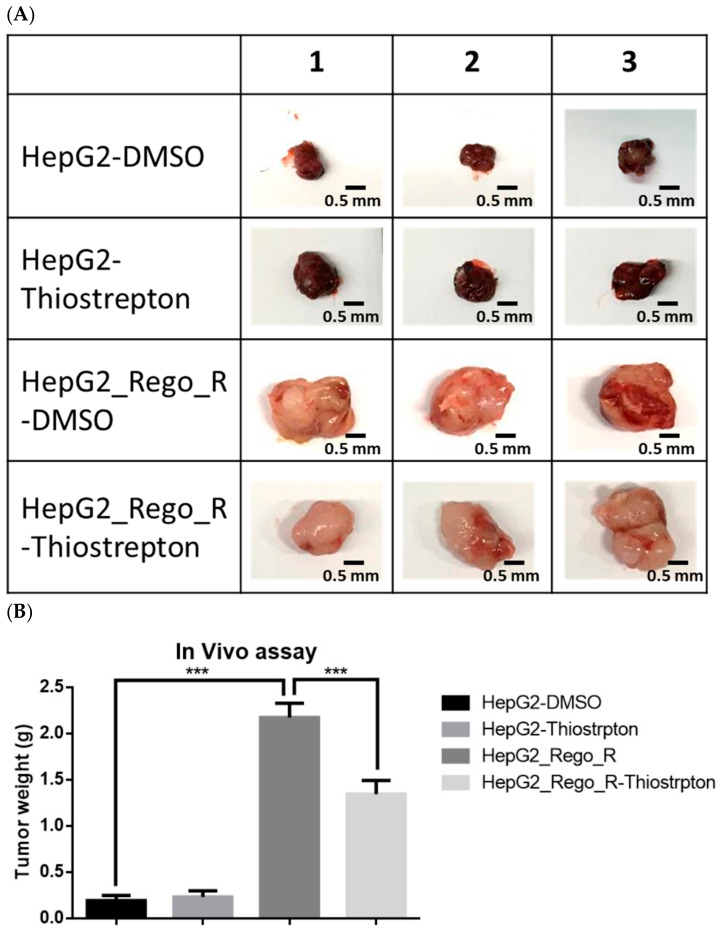
Comparative growths of mice xenografts with HepG2 and HepG2_Rego_R cells treated with or without thiostrepton or 0.05% DMSO control for 72 h as described in Materials and Methods. (**A**) Morphologies of the transplanted tumors and the size of each trial. Three representative trials are shown. (**B**) Comparisons of tumor weights of mice xenografts with HepG2 and HepG2_Rego_R cells treated with or without thiostrepton as described in Materials and Methods. Data are presented as mean ± SEM (n = 3) and were analyzed using one-way ANOVA and the Tukey post hoc test (*** *p* < 0.001). (**C**) Representative histochemical analysis of each tumor sample using hematoxylin staining as described in Materials and Methods. All are x 400 magnification. Necrosis is indicated using a red circle, giant cells using yellow arrows, and abnormal mitosis using a white arrow.

**Table 1 ijms-23-07782-t001:** Primer sequences of qRT-PCR and ChiP assay.

**qRT-PCR**	**Forward Primer**	**Reverse Primer**
SOX2	CAAAAATGGCCATGCAGGTT	AGTTGGGATCGAACAAAAGCTATT
ABCG2	TGGCTTAGACTCAAGCACAGC	GCCGGCAAGATGTGATGT
hGATA6	TACCACCTTATGGCGCAGAAA	AGGCTGTAGGTTGTGTTGTGG
SSEA4	ACAGCAGCCCCAGAAGTG	TGGAGTTTCAGGATTTGCAGT
CD44	GCAGTTTGCATTGCAGTCAAC	TCTGTCCTCCACAGCTCCATT
VIM	ACTGAGTACCGGAGACAGGT	GCAGCTTCAACGGCAAAGTT
hTWIST1	ATTCAGACCCTCAAGCTGGC	TCCATCCTCCAGACCGAGAA
ZEB1	GATGACCTGCCAACAGACCA	CCCCAGGATTTCTTGCCCTT
FOXM1	ATACGTGGATTGAGGACCACT	TCCAATGTCAAGTAGCGGTTG
AuroraA	GCTGGAGAGCTTAAAATTGCAG	TTTTGTAGGTCTCTTGGTATGTG
BIRC5	GCCCAGTGTTTCTTCTGCTT	CCGGACGAATGCTTTTTATG
**ChIP**	**Forward Primer**	**Reverse Primer**
SITE-1180	TTTCTGTGTAACTCACCAGGCAAG	TCTCCCATCTTTCCTACCCAGC
SITE-4714	GACTGTTTTGCTTGTGTTCCTTCC	GGTTTTACGCAGACCTTTGGAGG
SITE-16487	TACTTTCTGCTTTGTTTCGGGG	ACTGCCAAGGGATAACTCACTCC

## Data Availability

The raw data supporting the conclusions of this article will be made available by the authors, without reservations.

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
