# Peer review of "FOXM1-CD44 Signaling Is Critical for the Acquisition of Regorafenib Resistance in Human Liver Cancer Cells"

_ijms, 2022, doi:10.3390/ijms23147782_

Round 1

Reviewer 1 Report

The manuscript titled "FOXM1-CD44 Signaling Is Critical for the Acquisition of Regorafenib Resistance in Human Liver Cancer Cells", authored Kenly Wuputra and colleagues provides evidence on a FOXM1-CD44 related mechanism of chemoresistance in liver cancer. My Comments and Suggestions for Authors are the following:   1) The data provided by the authors are multilayered and convincing. The conclusions are clear and straightforward. Based on the fact that in vitro and ex vivo experiments and clinical analyses are provided, there are enough data to support publication of the aforementioned manuscript.   2) The manuscript is written in a comprehensive manner. However, there are are some minor points that should be addressed: -some spelling and grammatical errors throughout the text (for example line 100 "to form the spheroid" -> "to form spheroid") -Figure 3C and line 193. A representative picture before migration should be added. -Figure 6 and line 294. "CD44 promoter activity" is not presented in Figure 6, but of Figure 7. The authors should omit this phrase.   

Author Response

Prof. Dr. Benedetta Cinque

Prof. Dr. Paola Palumbo 

Guest editors of the International Journal of Molecular Sciences

Special issue on “Molecular Research on Cancer Stem Cell”

MDPI IJMS Editorial Office

St. Alban-Anlage 88, 4052 Basel

Switzerland

May 26th, 2022

Manuscript ID: ijms-1681703R1

Dear Editors,

We are pleased to resubmit our revised article entitled “FOXM1-CD44 Signaling Is Critical for the Acquisition of Regorafenib Resistance in Human Liver Cancer Cells” to the special issue on “Molecular Research on Cancer Stem Cell” in the International Journal of Molecular Sciences. We have revised the manuscript text to incorporate all suggestions and address the issues raised by reviewers 1 and 2 and add results of new experiments using shRNA and western blotting, sphere formation assay, CD44 promoter assay, and other experiments, such as viability assay. We hope that these revisions will be satisfactory to the reviewers.

Please find below our point-by-point responses to the reviewers’ comments.

[Reviewer 1]

My Comments and Suggestions for Authors are the following:   1) The data provided by the authors are multilayered and convincing. The conclusions are clear and straightforward. Based on the fact that in vitro and ex vivo experiments and clinical analyses are provided, there are enough data to support publication of the aforementioned manuscript.   2) The manuscript is written in a comprehensive manner. However, there are some minor points that should be addressed: -some spelling and grammatical errors throughout the text (for example line 100 "to form the spheroid" -> "to form spheroid") -Figure 3C and line 193. A representative picture before migration should be added. -Figure 6 and line 294. "CD44 promoter activity" is not presented in Figure 6, but of Figure 7. The authors should omit this phrase. 

Response: We have corrected the words as indicated, added the control figure, and corrected the incorrect figure number to Figure 7.

The contents of this manuscript have not been copyrighted or published previously. The contents of this manuscript are not now under consideration for publication elsewhere. All authors have directly participated in the planning, execution, or data analysis of the study. All authors of this paper have read and approved the final version submitted.

Thank you very much for your consideration of our manuscript.

Sincerely yours,

Kazunari K(Kazushige). Yokoyama, Ph.D.

Graduate Institute of Medicine

Kaohsiung Medical University

                                                        100 Shih-Chuan 1st Road, San Ming District

807 Kaohsiung, Taiwan

Phone:+886-7-312-1101, ext. 2729

Fax:+886-7-313-3849

e-mail: kazu@kmu.edu.tw

http://kazu.dlearn.kmu.edu.tw/

Reviewer 2 Report

In the paper entitled “FOXM1-CD44 signaling is critical for the acquisition of Regorafenib resistance in human liver cancer cells” Wuputra and co-workers have focused their attention on regorafenib resistance and on its underlying mechanisms, giving particular attention to FOXM1 and the CSC marker CD44.

Although the topic could be of interest, due to how the results were presented and discussed, this work requires additional more in-depth studies before being considered in any way for publication.

In order to help the authors to improve the article, here below are the main critical points under question that need to be addressed:

  • Line 98-99: In order to demonstrate the acquisition of resistance, it is necessary to perform a cell viability test on both parental and drug-selected cells treated with increasing concentration (2-10 uM) of Regorafenib and then calculate the relative IC50.
  • Line 123, Figure 1 legend: The title is not representative of the results showed into the figure. The expression of FOXM1 and CSC markers is not reported here but in figure 2
  • Line 133-136: In order to evaluate stem cell and CSC marker, the evaluation of mRNA expression is not satisfactory. It is necessary to evaluate the protein levels of these markers.
  • Line 230, Figure 4 legend: The title is not representative of the results showed in the figures: sphere formation is not reported here but in figure 5.
  • Line 241-245: sphere formation is evaluated after 6 days of treatment with the FOXM1 inhibitor (1uM). Which is the effects of this long-time exposure on cell viability? The reduced ability to form sphere might be due to the cytotoxic action of the inihibitor?
  • Figure 6: it is necessary to show the results obtained by not-target siRNA (Protein expression, sphere formation, CD44 promoter activity and cell viability).
  • Figure 6: CD-44 promoter activity is not not present here but it is reported in figure 7
  • Line 309: The author reported that FOXM1 knockdown impaired cell death, but results demonstrating this effect is not reported in the paper.
  • Line 367-402, In Vivo experiments: the injected cells for how many period were treated with the inhibitor? To demonstrate the role of FOXM1 it is necessary to treat the animals with the inhibitor and not only treat cells that will be injected.
  • Line 496-497: the half-maximal inhibitory concentration is not indicated in the text
  • In the Suppl. Material of Figure 3B, 2 blots of GAPDH were showed: 1 for TWIST, ZEB, CDH1 and CK18, and the other for CK7 and Vimentin. Instead, in Figure 3B there is only one GAPDH. Why?
  • In the Suppl. Material of Figure 6B, 3 blots of GAPDH were showed: 1 for FOXM1, SOX2 CD44m, AURKA in HepG2 cells; one for FOXM1 in Hep3B cells and one for SOX2, CD44, AURKA in Hep3B. Instead, in Figure 6B there are only two immunoblotting of  Why?
  • Finally, but not for importance, the quality of all immunoblots is very poor. It is necessary to try to improve it.

Author Response

Prof. Dr. Benedetta Cinque

Prof. Dr. Paola Palumbo 

Guest editors of the International Journal of Molecular Sciences

Special issue on “Molecular Research on Cancer Stem Cell”

MDPI IJMS Editorial Office

St. Alban-Anlage 88, 4052 Basel

Switzerland

May 26th, 2022

Manuscript ID: ijms-1681703R1

Dear Editors,

We are pleased to resubmit our revised article entitled “FOXM1-CD44 Signaling Is Critical for the Acquisition of Regorafenib Resistance in Human Liver Cancer Cells” to the special issue on “Molecular Research on Cancer Stem Cell” in the International Journal of Molecular Sciences. We have revised the manuscript text to incorporate all suggestions and address the issues raised by reviewers 1 and 2 and add results of new experiments using shRNA and western blotting, sphere formation assay, CD44 promoter assay, and other experiments, such as viability assay. We hope that these revisions will be satisfactory to the reviewers.

Please find below our point-by-point responses to the reviewers’ comments.

[Reviewer 2]

 In order to help the authors to improve the article, here below are the main critical points under question that need to be addressed:

  • Line 98-99: In order to demonstrate the acquisition of resistance, it is necessary to perform a cell viability test on both parental and drug-selected cells treated with increasing concentration (2-10 uM) of Regorafenib and then calculate the relative IC50.

Response: We have calculated the relative IC50 values of HepG2 and HepG2_Rego_R cells as well as Hep3B and Hep3B_Rego_R cells and summarized the findings in Supplementary Figure 1.

  • Line 123, Figure 1 legend: The title is not representative of the results showed into the figure. The expression of FOXM1 and CSC markers is not reported here but in figure 2.

Response: The title of Figure 1 was changed to “Cell morphologies, colony-forming abilities, and sphere-forming abilities of HepG2 and Hep3B parental cells and their regorafenib-resistant counterparts.” The expression of FOXM1 and CSC markers was moved to Figure 2 as suggested.

  • Line 133-136: In order to evaluate stem cell and CSC marker, the evaluation of mRNA expression is not satisfactory. It is necessary to evaluate the protein levels of these markers.

Response: We examined the protein expression of ABCG2, SOX2, CD44, and GATA6 besides that of SSEA4. Moreover, the antibody against SSEA4 can be used for immunostaining of the cells or fluorescence-activated cell sorting analysis. We present the western blotting results for these markers, except for SSEA4 and GATA6, in the revised manuscript.

  • Line 230, Figure 4 legend: The title is not representative of the results showed in the figures: sphere formation is not reported here but in figure 5.

Response: We have corrected the legends of Figures 4 and 5.

  • Line 241-245: sphere formation is evaluated after 6 days of treatment with the FOXM1 inhibitor (1uM). Which is the effects of this long-time exposure on cell viability? The reduced ability to form sphere might be due to the cytotoxic action of the inhibitor ?

Response: We examined the cell viability and sphere formation after a long treatment with FOXM1 inhibitor thiostrepton (for 6 days; Figure 5). To examine the cytotoxic effects of thiostrepton treatment, we performed the cell viability assay for HepG2 and HepG2_Rego_R cells after exposure to 1 mM thiostrepton. As shown in Figure S3 (upper panel), no difference was observed in cell viability between days 3 and 6. Thus, no large reduction in cell viability was noted after 6 days of thiostrepton treatment. Similar results were obtained in the case of Hep3B and Hep3B_Rego_R cells, which were also treated with 1 mM thiostrepton. To further confirm this conclusion, we used the shRNA vector to knock down the expression of FOXM1, and we obtained similar results (see Figure 6B, C). Thus, the sphere-forming ability was reduced specifically by the shRNA against FOXM1. We also obtained similar results for Hep3B and Hep3B_Rego_R cells (Figure S3). Therefore, the long-term exposure of the inhibitor did not affect the cell viability between HepG2 and HepG3 parental cells and their drug-resistant counterparts.

  • Figure 6: it is necessary to show the results obtained by not-target siRNA (Protein expression, sphere formation, CD44 promoter activity and cell viability).

Response: We agree with the reviewer. In Figure 6, the control data were derived from the off-target shRNA vector; we have revised the description of Figure 6A to add this information. The data for sphere formation in the case of shFOXM1 and control off-target shRNA were also added in Figure 6C, D. In addition, the cell viability and CD44 promoter activity of the cells treated with the off-target shRNA are now described in Figures 6 and 7.

  • Figure 6: CD-44 promoter activity is not present here but it is reported in figure 7

        Response: The CD44-promoter data have been added to Figure 7.

  • Line 309: The author reported that FOXM1 knockdown impaired cell death, but results demonstrating this effect is not reported in the paper.

Response: We have deleted the words “cell death” in the indicated sentence.

  • Line 367-402, In Vivo experiments: the injected cells for how many period were treated with the inhibitor? To demonstrate the role of FOXM1 it is necessary to treat the animals with the inhibitor and not only treat cells that will be injected.

(Response: We describe the treatment time-period (72 h) for thiostrepton and its control (0.05% DMSO) in the revised text, figure legends, and Materials and Methods.  

  • Line 496-497: the half-maximal inhibitory concentration is not indicated in the text

Response: We now describe the IC50 values in the revised text.

  • In the Suppl. Material of Figure 3B, 2 blots of GAPDH were showed: 1 for TWIST, ZEB, CDH1 and CK18, and the other for CK7 and Vimentin. Instead, in Figure 3B there is only one GAPDH. Why?

Response: We have added the control data for GAPDH in each case in the revised Suppl. Materials.

  • In the Suppl. Material of Figure 6B, 3 blots of GAPDH were showed: 1 for FOXM1, SOX2 CD44m, AURKA in HepG2 cells; one for FOXM1 in Hep3B cells and one for SOX2, CD44, AURKA in Hep3B. Instead, in Figure 6B there are only two immunoblotting of  Why?

Response: We have added the control data for GAPDH in each case in the revised Suppl. Materials.

  • Finally, but not for importance, the quality of all immunoblots is very poor. It is necessary to try to improve it.

Response: We have tried to select better-quality data from our western blotting data.

The contents of this manuscript have not been copyrighted or published previously. The contents of this manuscript are not now under consideration for publication elsewhere. All authors have directly participated in the planning, execution, or data analysis of the study. All authors of this paper have read and approved the final version submitted.

Thank you very much for your consideration of our manuscript.

Sincerely yours,

Kazunari K(Kazushige). Yokoyama, Ph.D.

Graduate Institute of Medicine

Kaohsiung Medical University

100 Shih-Chuan 1st Road, San Ming District

807 Kaohsiung, Taiwan

Phone:+886-7-312-1101, ext. 2729

Fax:+886-7-313-3849

e-mail: kazu@kmu.edu.tw

http://kazu.dlearn.kmu.edu.tw/ 

Round 2

Reviewer 2 Report

I thank the authors for taking my comments into consideration, however, one crucial point remains to be clarified.

1)      Line 367-402, In Vivo experiments: To demonstrate the role of FOXM1 it is necessary to treat the animals with the inhibitor and not only treat cells that will be injected. I understand that in vivo experiments cannot be done. Therefore, I suggest that the discussion might be revised in a manner more in keeping with the results shown.

Author Response

Prof. Dr. Benedetta Cinque

Prof. Dr. Paola Palumbo 

Guest editors of the International Journal of Molecular Sciences

Special issue on “Molecular Research on Cancer Stem Cell”

MDPI IJMS Editorial Office

St. Alban-Anlage 88, 4052 Basel

Switzerland

May 30th, 2022

Manuscript ID: ijms-1681703R1

Dear Editors,

We are pleased to resubmit our revised article entitled “FOXM1-CD44 Signaling Is Critical for the Acquisition of Regorafenib Resistance in Human Liver Cancer Cells” to the special issue on “Molecular Research on Cancer Stem Cell” in the International Journal of Molecular Sciences. We have revised the manuscript text to incorporate a suggestion and address the issues raised by reviewer 3. We hope that these revisions will be satisfactory to the reviewer 3.

Please find below our point-by-point responses to the reviewers’ comments.

[Reviewer 3]

  Line 367-402, In Vivo experiments: To demonstrate the role of FOXM1 it is necessary to treat the animals with the inhibitor and not only treat cells that will be injected. I understand that in vivo experiments cannot be done. Therefore, I suggest that the discussion might be revised in a manner more in keeping with the results shown.

[Answer]

Thank you very much for your kind suggestion. As suggested to the reviewer, we have revised the Discussion section to focus on the results only in in vivo xenotransplantation experiments (see the yellow parts).

In the in vivo model, mice transplanted with thiostrepton-treated regorafenib-resistant HepG2 cells showed smaller size of the xenotransplanted tumors derived from regorafenib-resistant HepG2 cells than did those transplanted with untreated regorafenib-resistant HepG2 cells (Figure 9A, B). To demonstrate the role of FOXM1 in vivo, we need to examine the effect of genetically induced depletion of FOXM1 gene or its inhibitor in the liver from regorafenib-resistant mice model. 

The contents of this manuscript have not been copyrighted or published previously. The contents of this manuscript are not now under consideration for publication elsewhere. All authors have directly participated in the planning, execution, or data analysis of the study. All authors of this paper have read and approved the final version submitted.

Thank you very much for your consideration of our manuscript.

Sincerely yours,

Kazunari K(Kazushige). Yokoyama, Ph.D.

Graduate Institute of Medicine

Kaohsiung Medical University

100 Shih-Chuan 1st Road, San Ming District

807 Kaohsiung, Taiwan

Phone:+886-7-312-1101, ext. 2729

Fax:+886-7-313-3849

e-mail: kazu@kmu.edu.tw   

http://kazu.dlearn.kmu.edu.tw/
